# Soil Chemical Properties Strongly Influence Distributions of Six Kalidium Species in Northwest China

Decheng Liu [1,†], Zongqiang Chang [2,†] , Xiaohui Liang [1] and Yuxia Wu [1,*]

1   State Key Laboratory of Herbage Improvement and Grassland Agro-Ecosystems, College of Ecology, Lanzhou University, Lanzhou 730000, China
2   Northwest Institute of Eco-Environment and Resources, Chinese Academy of Sciences, Lanzhou 730000, China
*   Correspondence: wuyx@lzu.edu.cn
†   These authors contributed equally to this work.

**Abstract:** The degrees of adaptive responses of different halophytes to saline–alkali soil vary substantially. *Kalidium* (Amaranthaceae), a genus comprised of six species of succulent euhalophytes with significantly differing distributions in China, provides ideal material for exploring the ecophysiological relationships involved in these variations. Thus, in a large-scale field survey in 2014–2018, samples of soil (at 20 cm depth intervals spanning 0 to 100 cm) and seeds were collected from areas where these six species are naturally distributed. Chemical properties of soils in the areas and germinability of the species' seeds in media with 0–500 mM NaCl and 0–250 mM $Na_2SO_4$ were then analyzed to test effects of salinity-related factors on the species' distributions. The pH of the soil samples mainly ranged between 8.5 and 10.5 and positively correlated with their mean total salt contents. Germination rates of all six species' seeds were negatively correlated with concentrations of NaCl and $Na_2SO_4$ in the media, and their recovery germination rates in distilled water were high (>74%). The results show that the species' distributions and chemical properties of their saline soils are strongly correlated, notably the dominant cation at all sites is $Na^+$, but the dominant anions at *K. cuspidatum* and *K. caspicum* sites are $Cl^-$ and $SO_4^{2-}$, respectively. Species-associated variations in concentrations of $Ca^{2+}$ were also detected. Thus, our results provide clear indications of major pedological determinants of the species' geographic ranges and strong genotype-environment interactions among *Kalidium* species.

**Keywords:** germination percentage; *Kalidium*; halophytes; pH; ion content; total salt contents; saline soil



## 1. Introduction

One of the diverse environmental factors that strongly affect terrestrial plants' natural distributions is soil salinity, which has growth-impairing and lethal effects on all plants when it exceeds species-dependent thresholds [1–3]. Further, salinity reportedly reduces crop yields on about a fifth of all irrigated land and, in combination with increasing global scarcity of water resources, salinization of soil and water is seriously threatening crop yields and future food production [4,5]. However, halophytes, comprising about 1% of the world's flora, can grow in saline environments with relatively high concentrations of electrolytes [6]. For example, the growth and development of glycophytes is severely inhibited by exposure to 100–200 mM of NaCl, while halophytes can tolerate and complete their life cycles at substantially higher concentrations [2,4,7]. This is due to anatomical adaptations in halophytes, such as salt bladders, salt hairs, and/or salt glands in the leaves [7,8] and various physiological tolerance mechanisms. For example, excess salt may be excreted through trichomes of halophytic grasses [8], or diluted by increases in the water content and thickness of succulent halophytes' leaves [9,10]. Moreover, different halophytes growing together on the same saline–alkali soil often have substantially differing elemental

concentrations, indicating that their physiological selectivity varies [11]. Generally, the dominant ions in salty habitats are $Na^+$ and $Cl^-$, but other ions (including $Ca^{2+}$, $Mg^{2+}$, K+, $SO_4^{2-}$, and $CO_3^{2-}$) may also be abundant [9,12]. Moreover, both their absolute and relative concentrations may vary, and influence the composition of the associated plant communities [13–15].

*Kalidium* (Amaranthaceae) is a genus of succulent halophytes with five species (*Kalidium caspicum* (L.) Ung.-Sternb., *Kalidium cuspidatum* (Ung.-Sternb.) Grubov., *Kalidium foliatum* (Pall.) Moq., *Kalidium gracile* Fenzl, and *Kalidium schrenkianum* Bunge ex Ung.-Sternb.), mainly distributed as shrubs in Northwest Asia and Southeast Europe. Some authorities have also recognized *K. wagenitzii* as an endemic species in Turkey, but others include it in *K. foliatum* [16]. In addition, two varieties of *K. cuspidatum* (var. *cuspidatum* and *sinicum* A. J. Li) have been recognized as separate species [17]. Here these varieties are called species, mainly due to genetic differences in DNA barcodes [17], but partly because our results indicate that they have significant adaptive differences. All six *Kalidium* species (including the two varieties of *K. cuspidatum* as species) naturally grow in deserts in northwest China. These succulent halophytes provide important fodder for livestock in winter, and have high ecological value for soil and water conservation in their semi-arid and arid areas. An extensive field survey showed that their distributions in China significantly differ (Figure 1). *K. foliatum* is the most widespread. Most *K. cuspidatum* sites are in Ningxia, *K. gracile* is mostly located in Gansu, Qinghai and Inner Mongolia. *K. sinicum* is naturally distributed in East Gansu and West Ningxia, while *K. caspicum* and *K. schrenkianum* are mainly restricted to regions north and south of the Tianshan Mountains in Xinjiang, respectively.

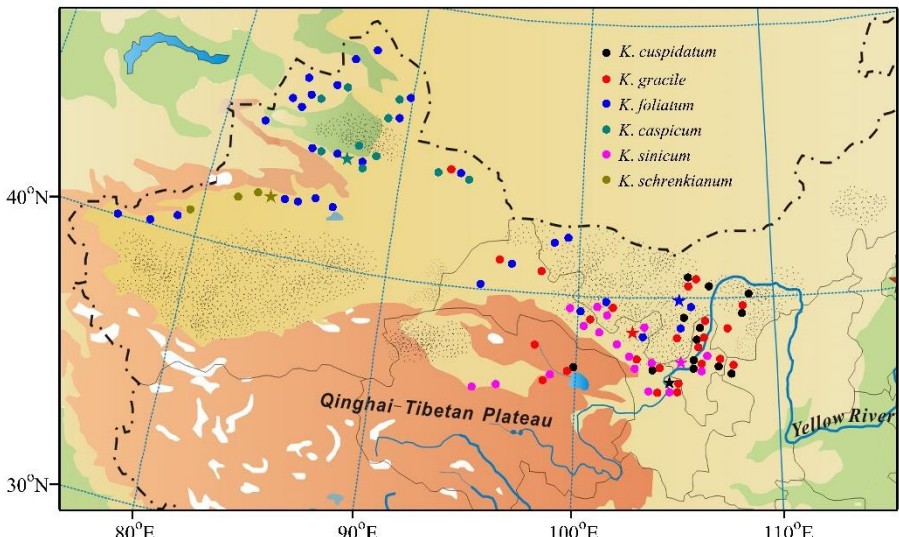

**Figure 1.** Map showing soil sampling sites in areas occupied by the six *Kalidium* species. Stars indicate seed sampling sites.

The adaptive evolution and origin of key halotolerance mechanisms have been intensively studied, as reviewed for example [2,18]. However, the NaCl concentration is not the sole stressor in saline environments. Thermal and water stresses are also often important [19,20]. Furthermore, salinity may be associated with extreme pH and/or variations in relative proportions of both cations and anions [21]. Effects of these variations have been less extensively studied, so this study focused on their impacts on distributions of the six *Kalidium* species in China. For this purpose, soil samples were collected from sites of the six species, across their ranges in China, then the pH, total salt contents, and ion contents of the soil were assayed at 20 cm depth intervals spanning 0 to 100 cm. In addition, the germinability of seeds of the six species was determined under different concentrations of NaCl and $Na_2SO_4$. Correlations between these abiotic factors and distributions of the

*Kalidium* species were then examined, to explore mechanisms affecting the relationship between biodiversity and ecosystem functions.

## 2. Materials and Methods

### 2.1. Field Investigation and Sampling

In a comprehensive field investigation of the areas where *Kalidium* Moq is distributed in northwest China during 2014–2018, we identified 103 representative sites, in total, of the six *Kalidium* species. These sites are mainly located in gravel deserts and/or gravel dunes, according to the FAO soil classification system (FAO 2016). Vegetation at the sites is dominated by *Kalidium* Moq and *Halocnemum strobilaceum* of the Amaranthaceae, *Halogeton arachnoideus* (Amaranthaceae), *Stipa glareosa* (Gramineae) and various other halophytes. Distances between neighboring sites mainly ranged from 150 to 200 km. Soil samples were collected from centers of these areas using a soil auger, with three replications (three-point sampling), at 20 cm depth intervals from 0 to 100 cm. Furthermore, mature seeds were collected from the six *Kalidium* species and stored in a refrigerator at $-20\,^{\circ}\text{C}$ before the start of experiments. The altitude and geographic coordinates of each site were measured using an Etrex GIS unit (Garmin, Taiwan). Locations of the collection sites are shown in Figure 1.

### 2.2. Determination of Soil Chemical Properties

The soil samples were air-dried, passed through a 1 mm sieve, then their pH was measured at a 1:5 soil: water ratio ($w/v$) using a PHS-25 pH meter (Shanghai Biocotek Inc., Shanghai, China), and their electrical conductivity (mS/cm) using a DDSJ-318 conductivity meter (Shanghai Biocotek Inc., Shanghai, China). Their contents of eight ions were also analyzed: $Na^+$, $K^+$, $Ca^{2+}$, and $Mg^{2+}$ using a 180-80 Polarized Zeeman atomic absorption spectrophotometer (Hitachi Inc., Tokyo, Japan); $CO_3^{2-}$ and $HCO_3^-$ by the double indicator titration method; $Cl^-$ by silver nitrate titration; and $SO_4^{2-}$ by the turbidimetric method [22].

### 2.3. Determination of Seeds' Germinability

Mean seed mass was calculated by weighing 1000 seeds of each *Kalidium* species with three replications (Table S1). Then germination and recovery experiments were conducted in an LRH 550-G programmed controlled-environment chamber (Shaoguan taihong, China) providing 16 h light/8 h dark cycles with cool white fluorescent lamps 100 μmol m$^{-2}$ s$^{-1}$ (Philips), and 25/19 $^{\circ}\text{C}$ day/night temperatures.

Seeds of the six *Kalidium* species were subjected to treatment with NaCl at seven concentrations by placing them on filter paper in Petri dishes (9 cm diameter) moistened with 10 mL of 0, 50, 100, 200, 300, 400, and 500 mM NaCl solution. Other batches were exposed to $Na_2SO_4$ with corresponding Na concentrations (0, 25, 50, 100, 150, 200 and 250 mM). Germination (regarded as emergence of the radicle from the seed by about 1 mm) was scored every day for 7 days. The germination rate at that point was calculated for each species, then ungerminated seeds were transferred to distilled water, and incubated under otherwise identical conditions. Germination of these seeds was scored for a further 7 days, after which the recovery germination percentage was calculated for each species.

### 2.4. Data Analysis

The soils' total salt contents were estimated from electrical conductivity measurements, using the following empirically determined linear relationship between NaCl concentration (y) and conductivity (x): y = 0.0159x (R$^2$ = 0.9811) [23]. Relationships between total salt contents and pH in 20 cm layers of the top 100 cm of soil at sites of the six species were examined by linear regression. The significance of differences in germination rates and recovery germination rates of seeds at different salt concentrations was tested by one-way ANOVA. Values of 18 bioclimatic factors covering most of the distributions of the six species were downloaded from the Global Climate Database (http://www.worldclim.org/bioclim, accessed on 9 November 2021). After excluding redundant bioclimatic factors, by applying a cumulative contribution ratio threshold of 80% [24], eight remained (Table S2). These were

used in combination with altitude and two soil factors (pH and total salt concentration of the soil) in the Principal Component Analysis (PCA) of abiotic factors affecting distributions of each of the six studied *Kalidium* species. For this, SPSS (Version 19.0., Chicago, IL, USA) was used (with the significance threshold set at $p < 0.05$).

## 3. Results

### 3.1. pH and Total Salt Contents

The pH of soils in the areas occupied by the *Kalidium* species mainly ranged from 8.5 to 10.5, although the *K. foliatum* sites had a wider range (7.3–11). The mean pH of soil samples from areas occupied by *K. cuspidatum*, *K. gracile*, *K. sinicum*, *K. foliatum*, *K. caspicum*, and *K. schrenkianum* was 9.41, 9.05, 9.13, 9.03, 8.94, and 8.90, respectively (Figure 2). Mean total salt contents in areas occupied by *K. cuspidatum*, *K. gracile*, *K. foliatum*, *K. caspicum* and *K. sinicum* were 22.4, 19.5, 17.6, 16.5 and 14.3 g/kg, respectively (Figure 2), while the highest mean value for any layer in *K. schrenkianum* areas was just 13.0 g/kg (in the 0–20 cm layer) and the overall mean was just 7.08 g/kg (Figures 2 and 3a). Generally, there was a positive correlation between pH values and mean salt contents in the topsoil (0–20 cm), but at >40 cm depths the relationship turned negative in areas occupied by all six species (Figure 3a). However, salt contents of samples from *K. cuspidatum* areas (with the highest mean pH and salt contents) significantly varied with depth while they were much more constant in *K. caspicum* areas (Figure 3). In addition, pH and total salt contents varied much more widely in soil samples from *K. foliatum*, *K. gracile*, and *K. sinicum* areas than in samples from *K. schrenkianum* areas (Figures 2 and 3).

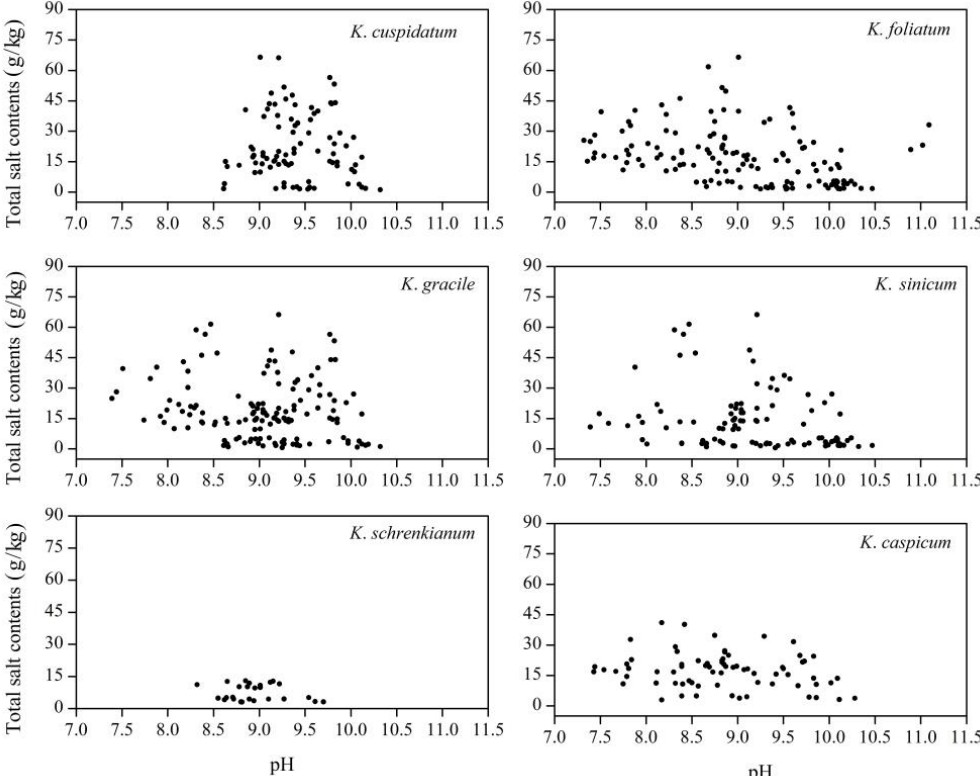

**Figure 2.** Scatter plots of total salt contents and pH of soil samples from areas occupied by indicated *Kalidium* species.

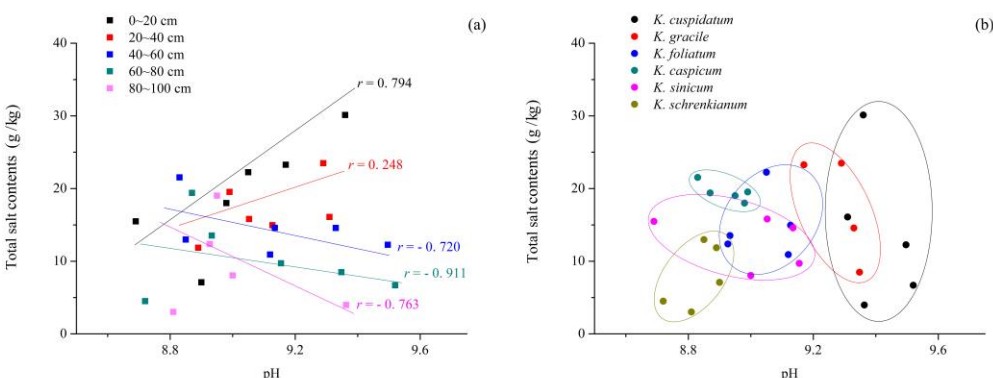

**Figure 3.** Linear regression plots showing the relationship between total salt contents and pH at indicated depths, and overall, in soil from areas occupied by the six *Kalidium* species. (**a**) total salt contents at different soil depth; (**b**) total salt contents at different species.

### 3.2. $Na^+$, $Ca^{2+}$, $K^+$, and $Mg^{2+}$ Concentrations

$Na^+$ concentrations in areas occupied by the six species were all high (Figure 4), especially in *K. cuspidatum* areas, where they declined with increases in depth but even at 80–100 cm exceeded 2.5 g/kg, the upper limit for sensitive crops according to the FAO. In *K. gracile* and *K. schrenkianum* areas, $Na^+$ concentrations peaked at 20–40 cm depth. Mean $Na^+$ concentrations in samples from all soil layers in areas occupied by the six species ranged from 0.90 to 4.22 g/kg. As shown in Table 1, $Na^+$ contents were also significantly positively correlated ($r > 0.89$) with total salt contents in soil from the six species' areas, except for 20–40 cm samples ($r = 0.632$), strongly suggesting that $Na^+$ made the largest contributions to the total salt contents.

**Table 1.** Correlation coefficient ($r$) between total salt contents and concentrations of indicated ions in soil from indicated depths in areas occupied by the six *Kalidium* species.

| Soil Depth | $Ca^{2+}$ | $Mg^{2+}$ | $K^+$ | $Na^+$ | $HCO_3^-$ | $Cl^-$ | $SO_4^{2-}$ |
|---|---|---|---|---|---|---|---|
| 0~20 cm | −0.841 * | 0.725 | 0.018 | 0.896 * | 0.992 ** | 0.856 * | 0.793 |
| 20~40cm | −0.093 | −0.156 | −0.680 | 0.632 | −0.969 ** | 0.689 | 0.031 |
| 40~60 cm | 0.677 | 0.980 ** | −0.140 | 0.986 ** | −0.809 | 0.880 * | 0.784 |
| 60~80 cm | 0.934 * | 0.991 ** | 0.567 | 0.972 ** | −0.873 | 0.933 * | 0.966 ** |
| 80~100 cm | 0.965 ** | −0.958 * | 0.660 | 0.973 ** | −0.502 | 0.167 | 0.344 |

*, $p < 0.05$; **, $p < 0.01$.

$Ca^{2+}$ was the second most abundant cation in all tested soil samples (Figure 4). In *K. foliatum* and *K. caspicum* areas, mean concentrations increased as soil depth increased, reaching 0.74 and 0.98 g/kg at 80–100 cm, respectively. In contrast, in *K. cuspidatum* and *K. sinicum* areas, mean $Ca^{2+}$ concentrations declined as depth increased and then increased (Figure 4), to <86 mg/kg at 40–60 cm in *K. cuspidatum* areas (Figure 4). Overall, as shown in Table 1, there was a negative correlation between mean $Ca^{2+}$ concentrations and total salt contents in the topsoil ($r = −0.841$, $p < 0.05$), but a positive correlation between them at 60–80 cm ($r = 0.934$, $p < 0.05$) and 80–100 cm ($r = 0.965$, $p < 0.01$).

$K^+$ concentrations did not exceed 260 mg/kg in any tested soil samples. In *K. gracile* and *K. sinicum* areas, as depth increased they first declined and then increased, and were almost twice as high in the former as the latter at each soil depth (0–60 cm) (Figure 4). $K^+$ concentrations significantly declined with depth (from 216, and 257 mg/kg, respectively, in topsoil) at all sites occupied by *K. schrenkianum* and *K. cuspidatum* (Figure 4).

$Mg^{2+}$ concentrations were also substantially lower than $Na^+$ and $Ca^{2+}$ concentrations (consistently < 300 mg/kg). In areas occupied by *K. caspicum*, *K. gracile*, and *K. sinicum* they first increased and then declined with increases in soil depth (Figure 4), peaking at 268 mg/kg at 40–60 cm in samples from *K. caspicum* areas (Figure 4). Overall, as shown in

Table 1, at sites of all six species, $Mg^{2+}$ concentrations were positively correlated with total salt contents at 40–60 cm ($r = 0.980$, $p < 0.01$) and 80–100 cm ($r = 0.991$, $p < 0.01$) depths.

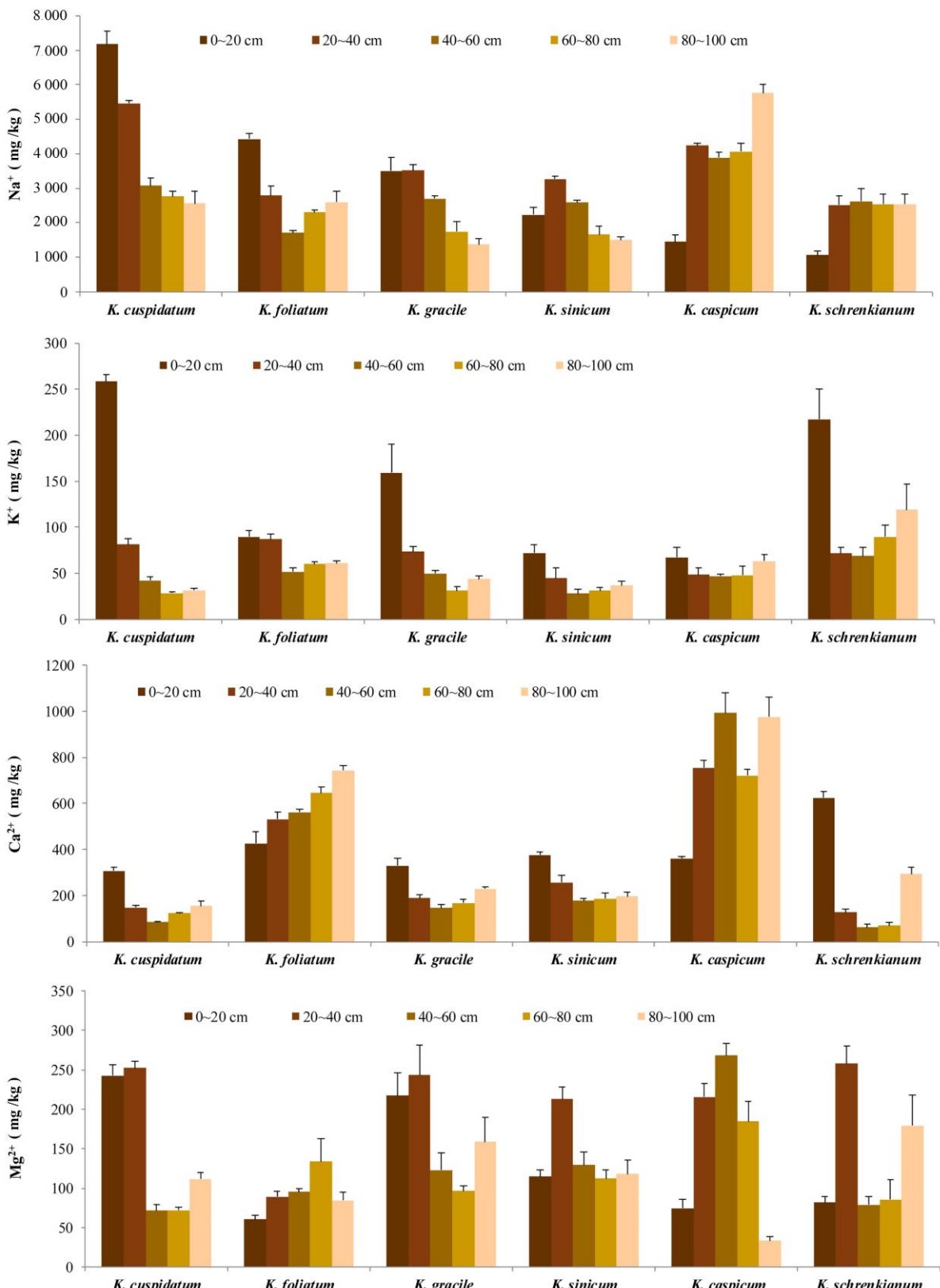

**Figure 4.** Concentrations of $Na^+$, $Ca^{2+}$, $K^+$, and $Mg^{2+}$ in soil from indicated depths in areas occupied by the six *Kalidium* species. The values shown are means with SE ($n = 3$).

### 3.3. Cl⁻, SO₄²⁻ and HCO₃⁻ Concentrations

Mean $Cl^-$ concentrations were highest in *K. cuspidatum* areas, where they declined from 10.62 g/kg in topsoil but still exceeded 2.5 g/kg at 80–100 cm depth, and lowest in *K. caspicum* areas, where the mean topsoil concentration was only 1.53 g/kg (Figure 5). It was also a major anion in *K. sinicum*, *K. foliatum* and *K. gracile* areas, where mean concentrations were 2.6, 2.7 and 3.2 g/kg, respectively (Figure 5). Overall, as shown in Table 1, $Cl^-$ concentrations were positively correlated with total salt contents at 0–20 cm ($r = 0.856$, $p < 0.05$), 40–60 cm ($r = 0.880$, $p < 0.05$) and 80–100 cm ($r = 0.933$, $p < 0.05$) depths.

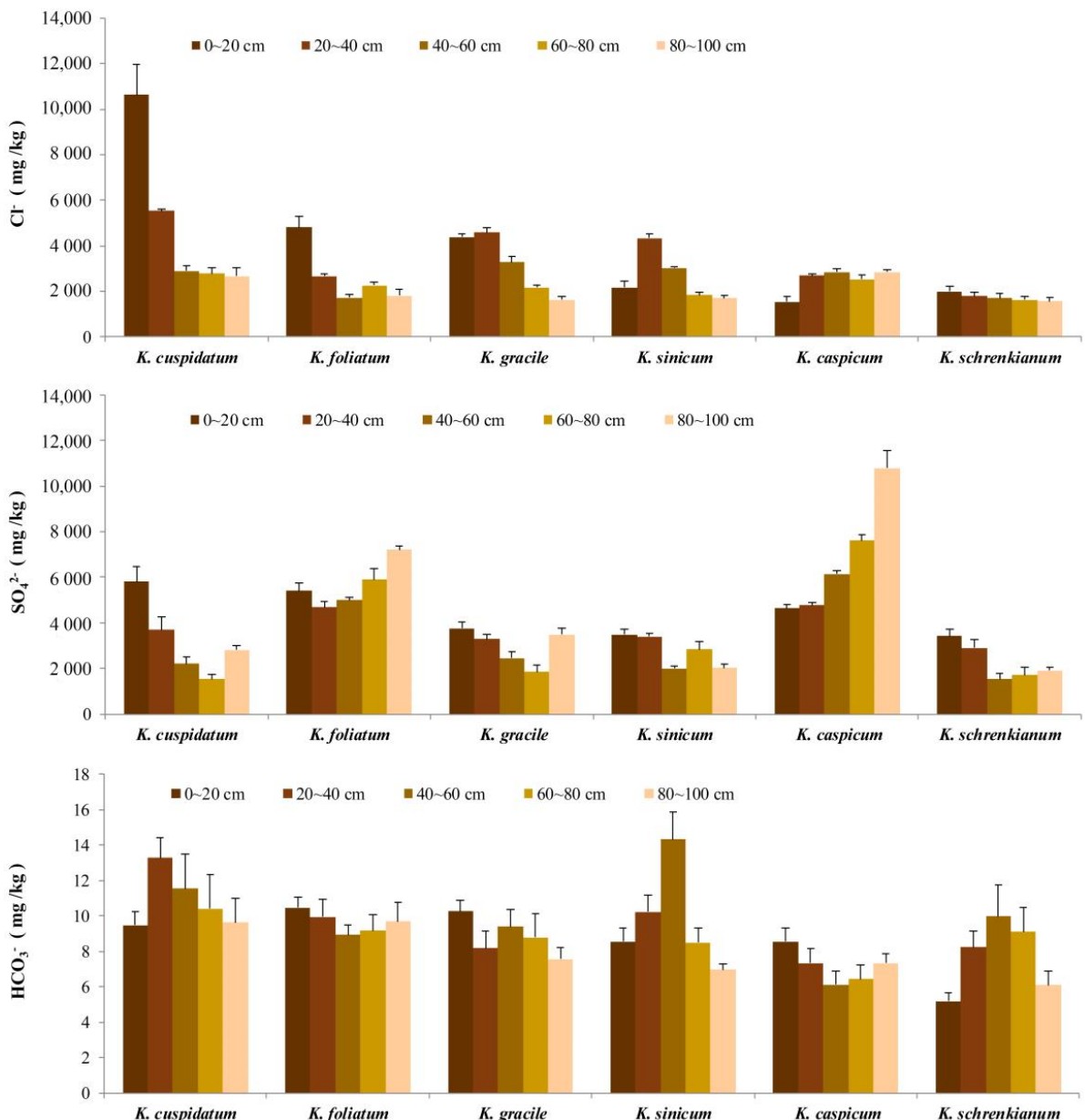

**Figure 5.** Concentrations of $Cl^-$, $SO_4^{2-}$, and $HCO_3^-$ in soil from indicated depths in areas occupied by the six *Kalidium* species. The values shown are means with SE ($n = 3$).

In *K. caspicum* areas, $SO_4^{2-}$ was the main anion, and its mean concentration increased as soil depth increased, reaching 11.78 g/kg at 80–100 cm, while concentrations were much lower in *K. cuspidatum* areas (just 1.52 g/kg at 60–80 cm depth) (Figure 5). In areas occupied by the other four species—*K. gracile*, *K. foliatum*, *K. sinicum*, and *K. schrenkianum*—the mean concentration first declined (from 3.7, 5.4, 3.5 and 3.43 g/kg, respectively, in topsoil) and

then increased as depth increased. Overall, it was only correlated with total salt contents at 60–80 cm depth (r = 0.966, *p* < 0.01) in areas occupied by the six *Kalidium* species (Figure 5, Table 1).

$HCO_3^-$ concentrations in soil from areas occupied by all six species were very low. Its mean concentrations first increased then declined as depth increased in *K. schrenkianum*, *K. cuspidatum* and *K. sinicum* areas (Figure 5), and were highest in the 40–60 cm layer of soil in *K. sinicum* areas, at just 14 mg/kg. Moreover, $HCO_3^-$ concentrations were negatively correlated with total salt contents at depths below 20 cm in areas occupied by all species (Figure 5, Table 1). $CO_3^{2-}$ was undetectable with the applied equipment in most samples.

### 3.4. Germination Rates

Germination rates of seeds of the six *Kalidium* species were negatively correlated with concentrations of both NaCl and $Na_2SO_4$ in the media (*p* < 0.05, Figures 6 and 7). In distilled water their germination rates ranged from 90.8% for *K. sinicum* to 100% for *K. caspicum*. However, less than half of all species' seeds germinated when the concentration exceeded 200 mM NaCl, and no *K. sinicum* seeds germinated at higher concentrations (400 or 500 mM NaCl) (Figure 6). Germination rates also declined with increases in $Na_2SO_4$ concentrations in the medium, and at 150 mM exceeded 50% (56.3%) for seeds of only one species (*K. caspicum*) (Figure 7).

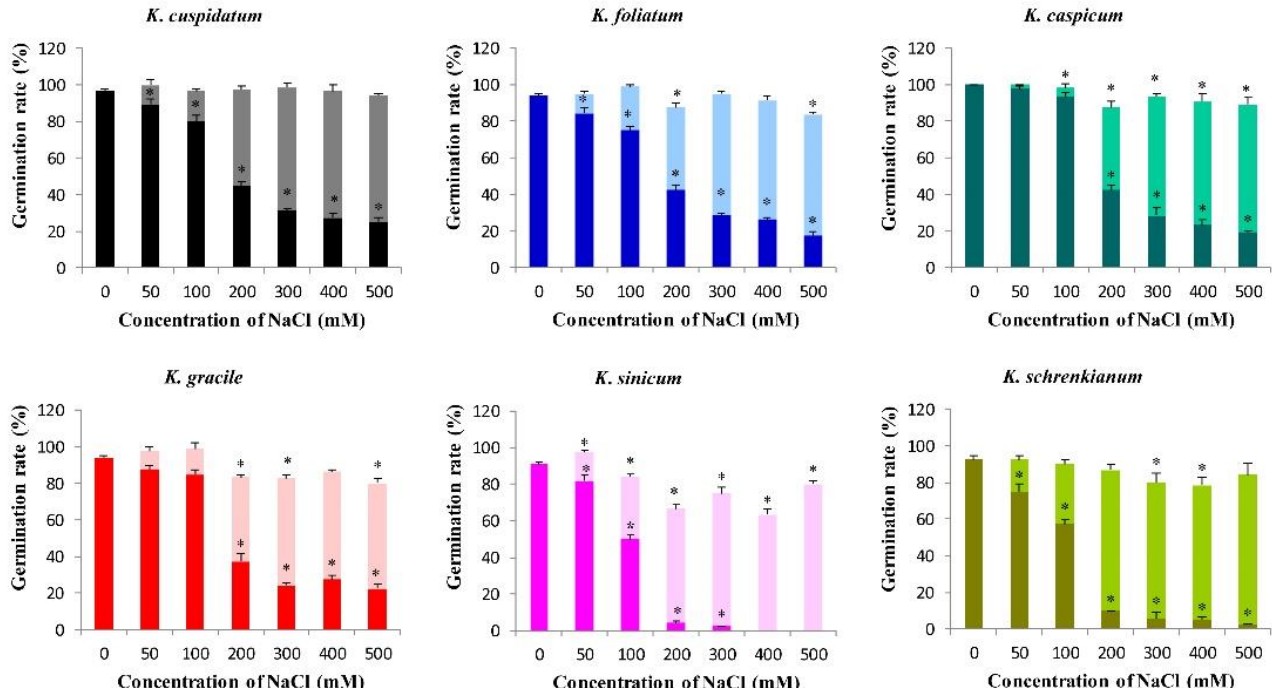

**Figure 6.** Total germination percentages after 14 days at the indicated NaCl concentrations in the six studied *Kalidium* species. Germination rates (%, means and standard deviations) of the six *Kalidium* species after treatments with indicated concentrations of NaCl (0, 50, 100, 200, 300, 400, and 500 mM NaCl) after 7 days. The dark bars indicate germination rates after the treatments and light bars the total percentages that germinated during the recovery treatment in distilled water with additional 7 days. Asterisks indicate significant differences in each condition respect to the corresponding control (according to Dunnet test, *p* < 0.05).

The recovery germination rates of seeds of all six *Kalidium* species after treatment with both salts were high. The recovery germination rates of *K. gracile* seeds following the NaCl treatment were negatively correlated with the NaCl concentration during the treatment, declining from 92.2 to 74.2% following exposure to 500 and 200 mM NaCl, respectively (Figure 6). However, the recovery germination rate of *K. sinicum* seeds exposed to 500 mM NaCl was high (>80%). Moreover, there were no significant differences in

recovery germination rates of *K. cuspidatum* seeds exposed to different NaCl concentrations ($p > 0.05$) and those of the other three species were all above 78% at 500 mM NaCl (Figure 6). Following treatment with $Na_2SO_4$, the recovery germination percentages of *K. sinicum* and *K. schrenkianum* seeds were lower than those of the other four species at the same concentrations, but were still high (76.6 and 83.2%, respectively), following exposure to the highest $Na_2SO_4$ concentration, 250 mM (Figure 7).

All six species produce small seeds (<0.5 g/1000 seeds, Table S1), and there was no linear relationship between the mean mass and germination rate of their seeds.

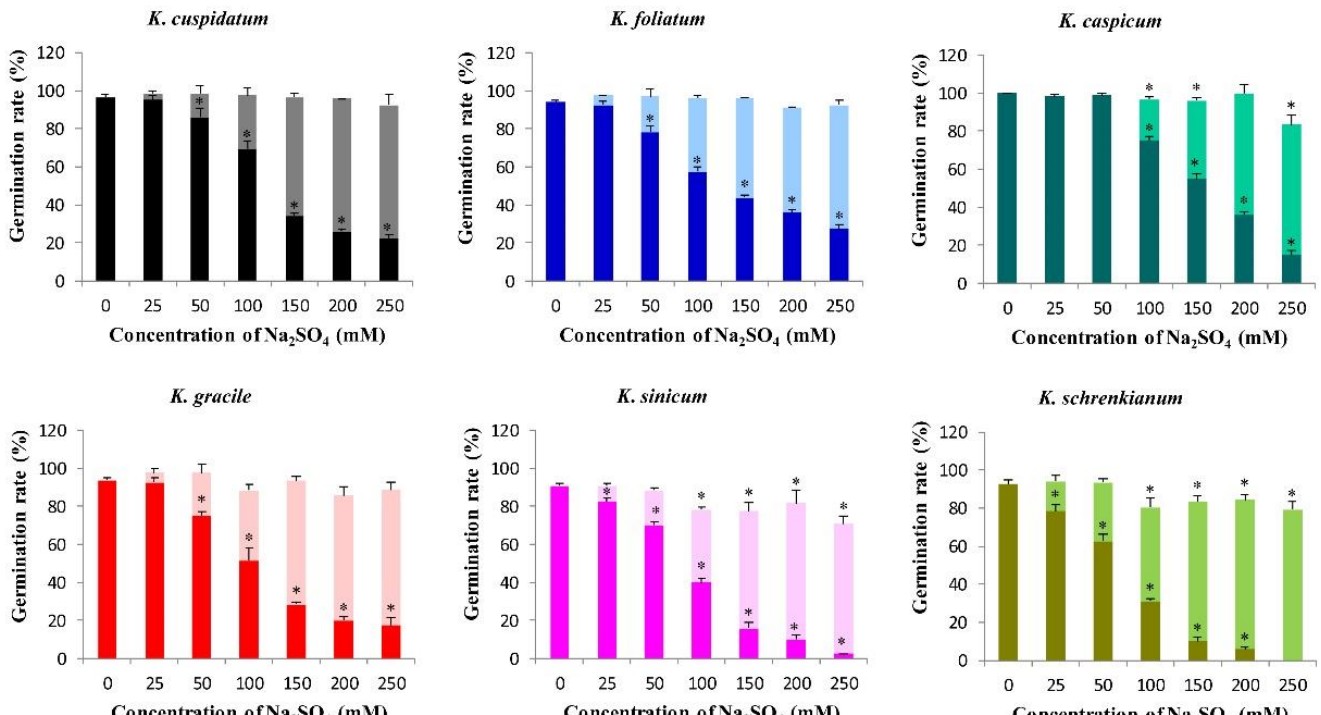

**Figure 7.** Total germination percentages after 14 days at the indicated $Na_2SO_4$ concentrations in the six studied *Kalidium* species. Germination rates (%, means and standard deviations) of the six *Kalidium* species after treatments with indicated concentrations of $Na_2SO_4$ (0, 25, 50, 100, 150, 200, and 250 mM) after 7 days. The dark bars indicate germination rates after the treatments and light bars the total percentages that germinated during the recovery treatment in distilled water with additional 7 days. Asterisks indicate significant differences in each condition respect to the corresponding control (according to Dunnet test, $p < 0.05$).

### 3.5. Principal Component Analyses

Two soil factors (pH and total salt contents), altitude, four temperature factors and two precipitation factors were used in PCA to explore relationships between abiotic factors and spatial distributions of each of the six *Kalidium* species (Figure 8 and Table S2). Principal Component (PC1) explained 40.4–50.5% of the variance and was most strongly influenced by temperature factors (maximum temperature of the warmest month, annual mean temperature, and mean temperature of the warmest quarter; Tables 2 and S2). PC2 explained 20.8–28.4% of the variance and largely reflected effects of precipitation factors on four of the six species. In addition, the two soil factors (total salt contents and pH) strongly contributed to PC3, explaining 9.96–18.4% of the variation in distribution of the six species. In total, PCs 1–3 explained >85% of the variation in the six species' distributions (Figure 8, Tables 2 and S2). The results strongly indicate that the most important ecological variables for adaptation to the species' saline environments are temperature during the driest month and precipitation. They also indicate that the selected ecological variables are strongly associated with the six *Kalidium* species' spatial distributions through their effects on the soil environment (Figure 1).

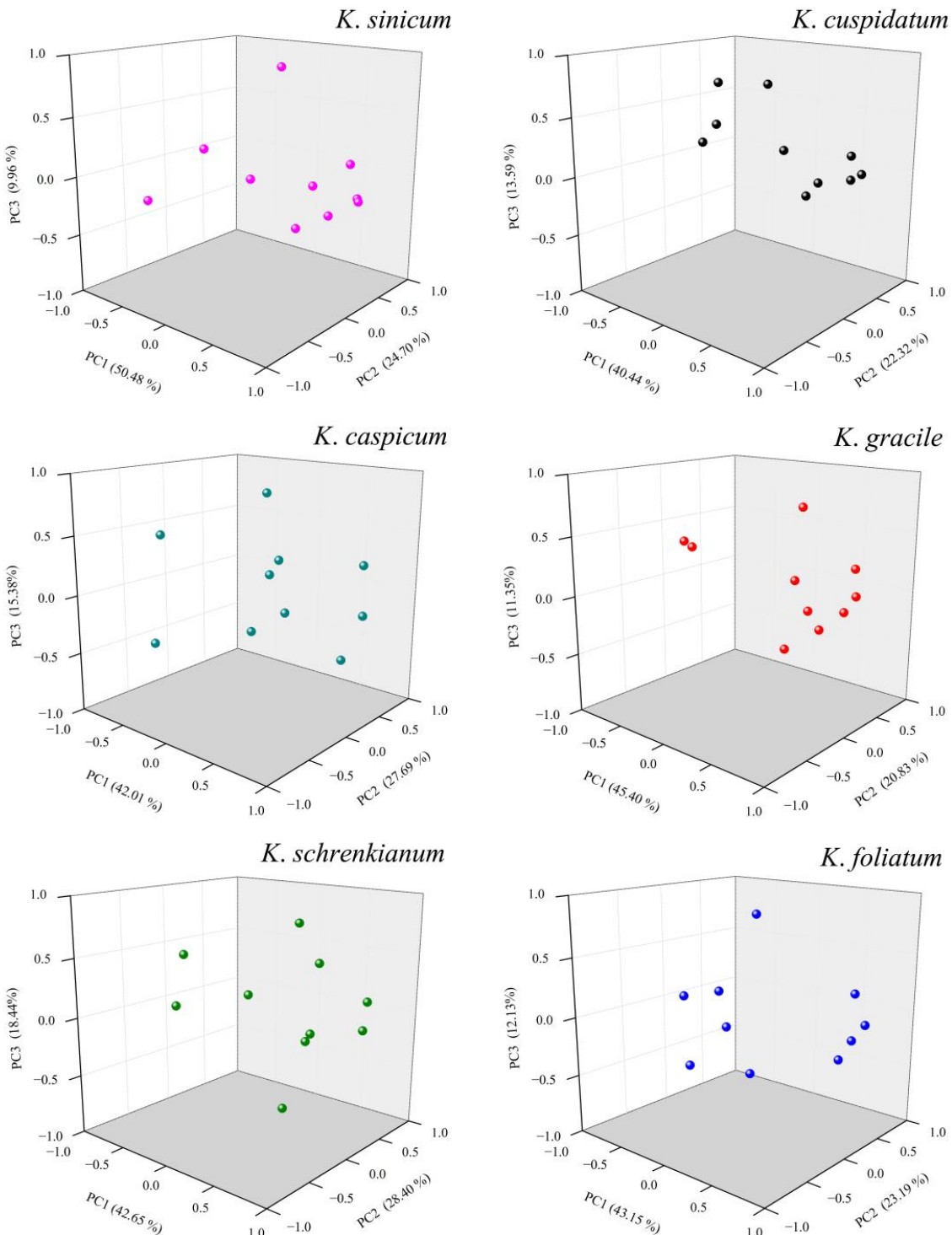

**Figure 8.** Score plots obtained from Principal Component Analysis (PCA) of effects of abiotic factors on distributions of the six *Kalidium* species. *x* axis, *y* axis, and *z* axis indicate the first, the second and the third Principle Component, respectively.

**Table 2.** Loadings of the main factors influencing the first three Principle Components obtained from Principle Component Analysis of the relationships between abiotic factors and distributions of the six *Kalidium* species.

| Species | Main Factors and Loadings (Correlation Coefficients) | | |
| --- | --- | --- | --- |
| | PC1 | PC2 | PC3 |
| *K. schrenkianum* | bio5 (0.971) | pH (0.797) | TS (0.719) |
| *K. sinicum* | bio9 (0.975) | bio14 (0.812) | TS (0.906) |
| *K.cuspidatum* | bio5 (0.892) | bio14 (0.707) | TS (0.766) |
| *K. capsicum* | bio1 (0.949) | bio14 (0.870) | pH (0.833) |
| *K. gracile* | bio1 (0.993) | bio12 (0.832) | bio14 (0.658) |
| *K. foliatum* | bio9 (0.924) | bio5 (0.713) | TS (0.840) |

bio1, annual mean temperature; bio5, max temperature of the warmest month; bio9, mean temperature of the driest quarter; bio12, annual average precipitation; bio14, precipitation of the driest month; TS, total salt contents.

## 4. Discussion

The mean pH was high (8.9–9.4) in soil samples from areas occupied by all six of the *Kalidium* species. Mean total salt contents were also high, but covered a substantial range (22.4, 19.5, 17.6, 16.5, 14.3, and 7.08 g/kg in soils from *K. cuspidatum*, *K. gracile*, *K. foliatum*, *K. caspicum*, *K. sinicum*, and *K. schrenkianum* areas, respectively). The indication that *K. schrenkianum* has relatively low salt tolerance, according to the low mean salt content of samples from areas it occupies, may at least partly explain why this species is restricted to a narrow range to the south of the Tianshan Mountains [18]. In areas occupied by all six species, total salt concentrations were positively related to pH at 0–20 cm and 20–40 cm soil depths ($r$ = 0.794 and $r$ = 0.248, respectively), indicating that salt contents are particularly strongly correlated to pH in topsoil in the study region.

In areas occupied by all the species, $Na^+$ was the main cation, and its concentrations were significantly correlated with total salt contents at all soil depths except 20–40 cm ($r$ = 0.632), showing that they either require high $Na^+$ concentrations for optimal growth and development or at least tolerate them [6]. $Ca^{2+}$ was the next most abundant cation at 0–40 cm soil depths, indicating that high concentrations of $Ca^{2+}$ and $Na^+$ likely accumulate in all the *Kalidium* species, as previously found in roots and leaves of *K. foliatum*, *K. cuspidatum,* and various other halophytes [25,26]. As a kind of antagonist, absorption of large amounts of $Ca^{2+}$ by roots could potentially alleviate damage to plants by other ions [27]. Concentrations of the major nutrient $K^+$, which is required by all living cells and often deficient in barren soil [28,29], ranged from 28.2 to 256.9 mg/kg in soil from *K. sinicum* and *K. cuspidatum* areas, respectively. Generally, in soils from areas of all six species the $K^+$ concentration was much lower than the $Na^+$ concentration and (hence) the $Na^+/K^+$ ratio was high (>30:1). In addition, there was no significant correlation between the $K^+$ concentration and total salt contents, indicating that the species' requirements for $Na^+$ and $K^+$ ions significantly differ. Mean $Mg^{2+}$ concentrations were highest and lowest in soil from *K. gracile* and *K. foliatum* areas (ca. 217 mg/kg and 2.3-fold lower, respectively), and substantially lower than concentrations of the other measured cations in all surveyed areas.

The most abundant anion was $Cl^-$ in *K. cuspidatum* areas (where concentrations of both $Na^+$ and $Cl^-$ were highest) and $SO_4^{2-}$ in *K. caspicum* areas (where $Cl^-$ concentrations were lowest). Thus, anions in soils in these areas were strongly dominated by $Cl^-$ and $SO_4^{2-}$, respectively (mainly balanced in both cases by $Na^+$). These were also the major ions in habitats of the other species, but there was less dominance by $Cl^-$ or $SO_4^{2-}$, e.g., mean $SO_4^{2-}$, $Cl^-$ and $Na^+$ concentrations in *K. foliatum* areas were 5.5, 2.6, and 2.8 g/kg, respectively. There were also wide variations in pH and total salt contents in soil samples from *K. foliatum* areas, indicating that the species has strong adaptive ability and, thus, can thrive in relatively diverse habits. Concentrations of $CO_3^{2-}$ were nondetectable and $HCO_3^-$ concentrations were very low (with no significant differences) in areas of the *Kalidium* species.

Plants must be sufficiently adapted to the salinity of their environments to germinate [19,30] and establish [31–34]. In our assays, the germination rates of seeds of the six *Kalidium* species were all negatively correlated with NaCl and $Na_2SO_4$ contents of the medium. Soil salinity fluctuates with precipitation, and can be alleviated in periods with high precipitation, so high proportions of seeds of many halophytes stored in highly saline soil may germinate during such periods [30,35,36]. Moreover, their recovery germination parameters may be major determinants of their distributions. The recovery germination rates of all six *Kalidium* species exceeded 74% after the NaCl and $Na_2SO_4$ treatments, corroborating the conclusions that the quality of topsoil is the first selective barrier affecting plants' distributions [14,35]. In addition, PCA showed that maximum temperature, summer rainfall and total salt contents of the soil strongly affect geographic distributions of the six *Kalidium* species. Similarly, distributions and yields of wild barley are clearly related to climatic factors, especially precipitation [37], and distributions of *Arabidopsis halleri* and *A. lyrate* are apparently linked to differences in their tolerance of the heavy metals Zn and Cd [38]. Clearly, therefore, ecological factors (and genetically-based adaptations to them) are key determinants of plants' distributions [18,37,38].

## 5. Conclusions

Adaptation to topsoil salinity in early stages is a major determinant of the six *Kalidium* species' geographic distributions in the study region. The dominant cation at all sites is $Na^+$, but the dominant anions at *K. caspicum* and *K. cuspidatum* sites are $SO4^{2-}$ and $Cl^-$, respectively. Both salinity and their distributions are affected by numerous interacting factors. Inter alia, temperatures during the driest month and precipitation directly and/or indirectly affect soils' salt contents and pH, which also strongly influence the six *Kalidium* species' distributions. Clearly, high tolerance of salinity stress is a key adaptive trait of halophytes, which has multiple evolutionary origins. Moreover, major changes have occurred in plants' distributions and population sizes during desertification, following which halophytes may occupy extensive semi-arid and arid regions.

**Supplementary Materials:** The following supporting information can be downloaded at: https://www.mdpi.com/article/10.3390/f13122178/s1:, Table S1: Masses of 1000 seeds of the six *Kalidium* species; Table S2: Loading matrix of the Principle Components (PCs).

**Author Contributions:** Y.W. and Z.C. conceived and managed the project; Y.W., D.L. and X.L. performed the experiments; Y.W., Z.C., D.L. and X.L. analyzed the data; Z.C. and D.L. performed statistical analyses; Y.W. and D.L. wrote the original manuscript; and Y.W. and Z.C. reviewed and edited the final manuscript. All authors have read and agreed to the published version of the manuscript.

**Funding:** This work was supported by the Central Government Guides Local Scientific and Technological Development Programs of Gansu Province (Grant Number 22ZY2QG001), National Key Research and Development Program of China (Grant Number 2022YFF1303301), the National Natural Science Foundation of China (NSFC, Grant Number 41871092), and Science and Technology Project of Forestry and Grassland Bureau of Gansu Province (Grant Number 2022kj063).

**Data Availability Statement:** The data presented in this study are available on request from the corresponding author.

**Acknowledgments:** We are grateful to Kuibing Meng and Fengzhu Zhang for collecting samples in the field.

**Conflicts of Interest:** The authors declare that they have no conflict of interest.

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
