# Peer review of "Soil Chemical Properties Strongly Influence Distributions of Six Kalidium Species in Northwest China"

_forests, doi:10.3390/f13122178_

Round 1

Reviewer 1 Report

The work is devoted to the study of 6 species of plants of the genus Kalidium growing in different regions of China. The authors of the work selected soil from different depths and seeds of the studied plants from different regions of China. The pH, anionic, cationic composition of the soil were analyzed. Next, using various statistical methods, correlations between abiotic factors and the distribution of Kalidium species were determined in order to study the mechanisms that affect the relationship between biodiversity and ecosystem functions.

The work is written very vaguely. There is no well-defined hypothesis. The results, discussions and conclusions are poorly described. Dear authors! Conclusions should be drawn by you yourself, based on your own results, and not take other people's reasoning, for example, page 12 lines 359-360. The article needs serious revision.

There are comments on the text.

For Foresters journal, literature in the text is given in [] and numbers.

Annotation page 1, line 12. Remove Moq family and put Amaranthaceae in brackets (Chenopodiaceae).

in the annotation line 14.  2014-2016, and in the methods 2014-2018 (page 3, line 94).

page 2, line 55.  Remove the word euphyllophytes or replace it with halophytes.

page 2, lines 55-56.  For species, write the full names of the authors, for example, Kalidium capsicum (L.) Ung.-Sternb.

page 2, line 68. Replace Kalidium foliatum with K. foliatum

page 2, line 90. What are the functions of the ecosystem, what does this mean?

Neither in the methods nor in the introduction is there a word about the regions of the study.

Why determine the germination of seeds?

Why were soils taken by horizons?

page 3,  line 116. Then germination and recovery experiments were conducted

What was restored and why?

page 5, line 173. For which sensitive crops? Researched halophytes.

page 6, line 189. For K. foliatum only.

page 6, line 195 ... as depth increased they first declined and then increased... What is meant.  Judging by the figure, there is no double increase.

Page 6, line 201. No increase, this is not a correct sentence. Why figure 3 if figure 4 is being discussed?

page 7, line 225.  Compared to what?

page 7,  line 225.  Its mean concentrations first increased then declined as depth increased. How much did it increase or decrease?

page 7 line 230. then why write about it?

page 8 line 234. Is this control?? There is no indication of control variants in the methods. In Figure 6, is zero a control?

page 9, line 253. What is the restorative germination of seeds? What was restored and why?

page 9, line 265. where table TS1. Why did you take the height above sea level parameter?

page 9, line 273. Where is the temperature? Where is the data on the factors you describe?

page 11, line 318. Where is this shown?

page 11, line 346. Why do you refer to wild barley?

page 12, line 358. Early stages of what?

page 12, line 354. What factors?

 page 12, line 355. This has not been proven in the work!

page 12, line 357. Many others are what? You didn't study it.

Figures 4, 8 are hard to read. In Figure 8, write in detail what the axes mean.

 In the text and figures, data are given in g/kg and mg/kg. Give the data in the same values.

Check the bibliography!

Author Response

Reviewer #1:

The work is devoted to the study of 6 species of plants of the genus Kalidium growing in different regions of China. The authors of the work selected soil from different depths and seeds of the studied plants from different regions of China. The pH, anionic, cationic composition of the soil were analyzed. Next, using various statistical methods, correlations between abiotic factors and the distribution of Kalidium species were determined in order to study the mechanisms that affect the relationship between biodiversity and ecosystem functions.

The work is written very vaguely. There is no well-defined hypothesis. The results, discussions and conclusions are poorly described. Dear authors! Conclusions should be drawn by you yourself, based on your own results, and not take other people's reasoning, for example, page 12 lines 359-360. The article needs serious revision.

 We have revised the manuscript according to your suggestions.

For Foresters journal, literature in the text is given in [ ] and numbers.

We have changed all the literatures in revised ms in [ ] and numbers.

Annotation page 1, line 12. Remove Moq family and put Amaranthaceae in brackets (Chenopodiaceae).

We have verified it from the new edition “Flora of China” (https://www.cvh.ac.cn/species/taxon_tree.php), and put Amaranthaceae in brackets (Chenopodiaceae) (page 1, line 12).

in the annotation line 14.  2014-2016, and in the methods 2014-2018 (page 3, line 94).

We have verified this field survey in 2014-2018 in revised ms (page 1, line 14 and page 2, line 87)

page 2, line 55.  Remove the word euphyllophytes or replace it with halophytes.

We have replace he word euphyllophytes with halophytes in revised ms (page 2, line 52)

page 2, lines 55-56.  For species, write the full names of the authors, for example, Kalidium capsicum (L.) Ung.-Sternb.

We have write the full names of six species with the full name of the authors in revised ms (lines 52-55)

page 2, line 68. Replace Kalidium foliatum with K. foliatum

We have replaced Kalidium foliatum with K. foliatum (page 2, line 65)

page 2, line 90. What are the functions of the ecosystem, what does this mean?

Neither in the methods nor in the introduction is there a word about the regions of the study.

Sampling sites were chosen as following: we conducted a comprehensive field investigation of the areas where Kalidium distributed in Northwest China during 2014-2018, finally 103 sites which could represent all distributions of six Kalidium species were collected for the study. And these sites were mainly located in the center of saline deserts or gravel deserts where dominated by Kalidium (Amaranthaceae), Halocnemum strobilaceum (Amaranthaceae), Halogeton arachnoideus (Amaranthaceae), Stipa glareosa (Gramineae) or some other halophytes, the distances between sites mainly ranged from 150 km to 200 km (pages 2-3, lines 86-92).

Why determine the germination of seeds?

Germination is an important stage in the life cycle of species growing in saline environments.

Why were soils taken by horizons?

For annual plants, because most of their root growth may be confined to top soil, 0-25cm soil root zone will be enough, but for the genus Kalidium, as perennial shrubs, most of them are distributed in the desert, and the roots are elongated in depth to absorb water and nutrients. At the same time, the salt content in desert environment will return to the soil with rainfall, so the analysis of soil depth of 0-100cm will be more comprehensive.

page 3,  line 116. Then germination and recovery experiments were conducted

What was restored and why?

In the natural environment, the seeds of most halophytes, in order to resist salt stress, usually remain dormant in the surface layer where the salt concentration is much higher than that of the underlying soil. The increase of salt stress can not only reduce the seed germination rate and delay the seed germination process, but also completely inhibit the germination process when the salinity exceeds the tolerance range of the species. When the stress conditions were alleviated, most halophytes showed an important process of resuming germination. This may have important ecological significance for plants growing in high salinity environment, and reflect a physiological response in the process of species evolution under strong selection pressure.

page 5, line 173. For which sensitive crops? Researched halophytes.

sensitive crops(these including the planting crops belong to Panicoideae,Triticeae, Oryzinae, Asteridae, Fabids, Maivids and so on)

page 6, line 189. For K. foliatum only.

Both K. cuspidatum and K. sinicum distributed areas.

page 6, line 195 ... as depth increased they first declined and then increased... What is meant.  Judging by the figure, there is no double increase.

In K. gracile and K. sinicum areas, and were almost twice as high in K. gracile as K. sinicum at each soil depth (0-60 cm).

Page 6, line 201. No increase, this is not a correct sentence. Why figure 3 if figure 4 is being discussed?

page 7, line 225.  Compared to what?

And we checked the text and according figures, figure number in lines 202-203 is error, And we have corrected the figure 4 in the revised ms (page 6, lines 198-199).

page 7,  line 225.  Its mean concentrations first increased then declined as depth increased. How much did it increase or decrease?

Its mean concentrations first increased then declined as depth increased in K. schrenkianum (line 222).

page 7 line 230. then why write about it?

To indicate that CO32- as anion almost can detected in the genus Kalidium distributed areas.

page 8 line 234. Is this control?? There is no indication of control variants in the methods. In Figure 6, is zero a control?

distilled water as control

page 9, line 253. What is the restorative germination of seeds? What was restored and why?

The germination rate at that point was calculated for each species, then ungerminated seeds were transferred to distilled water, and incubated under identical conditions.

page 9, line 265. where table TS1. Why did you take the height above sea level parameter?

table TS1 has been in Supplementary File.

We were confused by “Why did you take the height above sea level parameter?”

page 9, line 273. Where is the temperature? Where is the data on the factors you describe?

the temperature is located from “Materials and Methods”as following:

Values of 18 bioclimatic factors covering most of the distributions of the six species were downloaded from the Global Climate Database

(http://www.worldclim.org/bioclim). page 4, lines 133-135

page 11, line 318. Where is this shown?

This is shown in table1 (page 7, lines 180-182).

page 11, line 346. Why do you refer to wild barley?

Correlation tests returned significant correlations with environmental variables that relate to water availability from 178 wild barley accessions from eight climatically divergent sites. And to indicate that ecological factors (and genetically-based adaptations to them) are key determinants of plants distributions.

page 12, line 358. Early stages of what?

We can’t find Early stages in according to the ms (page 12, line 358).

And we have corrected the expression in the revised ms (page 13, lines 351-359).

page 12, line 354. What factors?

Factors such as salt contents and pH, temperatures during the driest month, precipitation and so on.

 page 12, line 355. This has not been proven in the work!

We have changed this expression in the revised ms (page 13, lines 351-359).

page 12, line 357. Many others are what? You didn't study it.

And we have corrected the expression in the revised ms (page 13, lines 351-359).

Figures 4, 8 are hard to read. In Figure 8, write in detail what the axes mean.

We have added all details under each figure in the revised ms (Figures 4-8).

 In the text and figures, data are given in g/kg and mg/kg. Give the data in the same values.

The order of magnitude of total salt content and element content in soil is different, so different units are used.

Check the bibliography!

We have checked all the bibliography in the revised ms.

Reviewer 2 Report

Soil chemical properties strongly influence distributions of six Kalidium species in Northwest China

I read this MS with great interest. In this research soil samples were collected from sites of the six species, across their ranges in China, then the pH, total salt contents, and ion contents of the soil were assayed, at 20-cm depth intervals spanning 0 to 100 cm. In addition, the germinability of seeds of the six species was determined under different concentrations of NaCl and Na2SO4. Correlations between these abiotic factors and distributions of the Kalidium species were then examined, to explore mechanisms affecting the relationship between biodiversity and ecosystem functions. The results showed that the pH of the soil samples positively correlated with their mean total salt contents. Germination rates of all six species’ seeds were negatively correlated with concentrations of NaCl and Na2SO4 in the media, and their recovery germination rates in distilled water were high (> 74%). The results show that the species’ distributions and chemical properties of their saline soils are strongly correlated, notably the dominant cation at all sites is Na+, but the dominant anions at K. cuspidatum and K. caspicum sites are Cl- and SO42-, respectively. The results provide clear indications of major pedological determinants of the species’ geographic ranges and strong genotype-environment in-teractions among Kalidium species.

The MS is clearly written but some comments and suggestions should be considered to improve the quality of the manuscript. These comments are as follows:

-         Lines 14. The authors reported that this field survey in 2014-2016 while in Line 94 they mentioned different period 2014-2018. Which is true?

-         I recommend to transfer Fig.1 from introduction to Materials and methods.

-         The citing references format throughout the MS is different from journal style. Please be sure and use numerical system.

-         Line 102. Why the seeds of six Kalidium species stored in a refrigerator

at -20 ℃?

-         Line 118. Please provide the light intensity.

-         Line 146. The authors presented the effect of soil chemical analysis on the distribution of six Kalidium species. But what about the physical analysis of the six sites? Is it the same for all? If it is differed from one site to another, where its effect?

-         In Figs. 4-7. What the authors mean by the bars above the columns? Are they referring to SD or SE. Please clarify. Additionally, it would better to put the letters above the columns to show the significant difference.

-         In all Figure captions. Each Fig. must be self-explanatory. Please add all details under each figure. For example, Figs 6 and 7 please add the germination conditions and period.

-         Line 355. Please change the font of (Inter alia) to normal not bold.

-         Line 360. The discussion part should not contain references.

-         Lines 334-335. Please modify this sentence as follows:

-         Plants must be sufficiently adapted to the salinity of their environments to germinate (Ungar 1978; Tobe et al. 2000) and establish (Attia et al., 2020; Hassan et al., 2021; Yasir et al., 2021; Hassan et al., 2021).

Attia et al., 2020. Induced anti-oxidation efficiency and others by salt stress in Rosa damascena Miller. Scientia Horticulturae 274, 2020 , 109681. 10.1016/j.scienta.2020.109681

Yasir et al., 2021. Exogenous sodium nitroprusside mitigates salt stress in lentil (Lens culinaris medik.) by affecting the growth, yield, and biochemical properties. Molecules 26, 9, 2576. 10.3390/molecules26092576

 Hassan et al., 2021. Mitigation of salt-stress effects by moringa leaf extract or salicylic acid through motivating antioxidant machinery in damask rose. Canadian Journal of Plant ScienceVolume 101, 2, 157 – 165. 10.1139/cjps-2020-0127.

Hassan et al., 2021. Chitosan nanoparticles effectively combat salinity stress by enhancing antioxidant activity and alkaloid biosynthesis in Catharanthus roseus (L.) G. Don. Plant Physiology and Biochemistry 162, 291 – 300. https://doi.org/10.1016/j.plaphy.2021.03.004

-         Line 233. lease revise the beginning of the sentence (Error! Reference source not found.) and re-edit it again.

-         References. I noticed that all references are not up to date. I recommend authors to use some recent references related to the article instead of the old ones.

Author Response

Reviewer #2:

I read this MS with great interest. In this research soil samples were collected from sites of the six species, across their ranges in China, then the pH, total salt contents, and ion contents of the soil were assayed, at 20-cm depth intervals spanning 0 to 100 cm. In addition, the germinability of seeds of the six species was determined under different concentrations of NaCl and Na2SO4. Correlations between these abiotic factors and distributions of the Kalidium species were then examined, to explore mechanisms affecting the relationship between biodiversity and ecosystem functions. The results showed that the pH of the soil samples positively correlated with their mean total salt contents. Germination rates of all six species’ seeds were negatively correlated with concentrations of NaCl and Na2SO4 in the media, and their recovery germination rates in distilled water were high (> 74%). The results show that the species’ distributions and chemical properties of their saline soils are strongly correlated, notably the dominant cation at all sites is Na+, but the dominant anions at K. cuspidatum and K. caspicum sites are Cl- and SO42-, respectively. The results provide clear indications of major pedological determinants of the species’ geographic ranges and strong genotype-environment interactions among Kalidium species.

The MS is clearly written but some comments and suggestions should be considered to improve the quality of the manuscript. These comments are as follows:

-         Lines 14. The authors reported that this field survey in 2014-2016 while in Line 94 they mentioned different period 2014-2018. Which is true?

We verified this field survey in 2014-2018 in the revised MS (page 2, line 87).

-         I recommend to transfer Fig.1 from introduction to Materials and methods.

We have transferred Fig.1 from introduction to Materials and methods (page 3, lines 99-103).

-         The citing references format throughout the MS is different from journal style. Please be sure and use numerical system.

We have cited references format throughout the revised MS according to the journal style.

-         Line 102. Why the seeds of six Kalidium species stored in a refrigerator at -20 ℃?

In a high-temperature and humid environment, the seeds have strong respiration and consume more organic substances, so the germination rate is low when planting. Therefore, in order to maintain the germination rate, dry seeds need to be stored reasonably. The low temperature, dry storage environment can just reduce the respiration of seeds, so the seeds will be stored in such an environment.

-         Line 118. Please provide the light intensity.

We provide the light intensity 100 μmol m-2 s-1 in the revised MS (page3, Lines 115-116)

-         Line 146. The authors presented the effect of soil chemical analysis on the distribution of six Kalidium species. But what about the physical analysis of the six sites? Is it the same for all? If it is differed from one site to another, where its effect?

We have analyzed physical factors, values of 18 bioclimatic factors covering most of the distributions of the six species were downloaded from the Global Climate Database (http://www.worldclim.org/bioclim) (page 4, lines 133-135). And the results shown in Figure 8 and Table2.

-         In Figs. 4-7. What the authors mean by the bars above the columns? Are they referring to SD or SE. Please clarify. Additionally, it would better to put the letters above the columns to show the significant difference.

The values shown are means with SE (n = 3). If we put Different lowercase letters above the bars indicate significant differences between depths ( 0-100cm) for each species, and different capital letters indicate significant differences between species for each depths ( 0-100cm) according to Tukey’s test (α = 0.05). This will cause the diagram to be very confusing. For the sake of clarity and simplification, we have omitted this step.

-         In all Figure captions. Each Fig. must be self-explanatory. Please add all details under each figure. For example, Figs 6 and 7 please add the germination conditions and period.

We have added all details under each figure in the revised MS.

-         Line 355. Please change the font of (Inter alia) to normal not bold.

We have changed the font of (Inter alia) to normal not bold in the revised MS.

-         Line 360. The discussion part should not contain references.

We have deleted references in the conclusions part in the revised MS (Line 360).

-         Lines 334-335. Please modify this sentence as follows:

We have modified this sentence as follows in the revised MS (Lines 334-335)

-         Plants must be sufficiently adapted to the salinity of their environments to germinate (Ungar 1978; Tobe et al. 2000) and establish (Attia et al., 2020; Hassan et al., 2021; Yasir et al., 2021; Hassan et al., 2021).

Attia et al., 2020. Induced anti-oxidation efficiency and others by salt stress in Rosa damascena Miller. Scientia Horticulturae 274, 2020 , 109681. 10.1016/j.scienta.2020.109681

Yasir et al., 2021. Exogenous sodium nitroprusside mitigates salt stress in lentil (Lens culinaris medik.) by affecting the growth, yield, and biochemical properties. Molecules 26, 9, 2576. 10.3390/molecules26092576

 Hassan et al., 2021. Mitigation of salt-stress effects by moringa leaf extract or salicylic acid through motivating antioxidant machinery in damask rose. Canadian Journal of Plant ScienceVolume 101, 2, 157 – 165. 10.1139/cjps-2020-0127.

Hassan et al., 2021. Chitosan nanoparticles effectively combat salinity stress by enhancing antioxidant activity and alkaloid biosynthesis in Catharanthus roseus (L.) G. Don. Plant Physiology and Biochemistry 162, 291 – 300. https://doi.org/10.1016/j.plaphy.2021.03.004

We have added all the above list references in the revised MS (page 15 Lines 430-439)

-         Line 233. lease revise the beginning of the sentence (Error! Reference source not found.) and re-edit it again.

Germination rates of seeds of the six Kalidium species were negatively correlated with (Line 233) we can’t find Error, maybe the lines mis-leading?

-         References. I noticed that all references are not up to date. I recommend authors to use some recent references related to the article instead of the old ones.

We have added some recent references related to the article and delete the old ones in the revised MS.

Reviewer 3 Report

Manuscript Number: forests-2079348

Title: Soil chemical properties strongly influence distributions of six Kalidium species in Northwest China

FORESTS

The manuscript entitled Soil chemical properties strongly influence distributions of six Kalidium species in Northwest China» provide clear indications of major
pedological determinants of the species’ geographic ranges and strong genotype-environment in teractions among Kalidium species. Some of the results are interesting and the objectives are clear. But, the discussion is not appropriated y basically repeat the results.

The manuscript should be edited by a native scientific English speaker.

Author Response

Reviewer #3:

The manuscript entitled “Soil chemical properties strongly influence distributions of six Kalidium species in Northwest China» provide clear indications of major pedological determinants of the species’ geographic ranges and strong genotype-environment interactions among Kalidium species. Some of the results are interesting and the objectives are clear. But, the discussion is not appropriated y basically repeat the results.

The manuscript should be edited by a native scientific English speaker.

    We have corrected our manuscript and improve the discussion.

In addition, we have our manuscript professionally edited by Sees-editing.co.uk, polished and checked all English expression before submitted to Forests.

Round 2

Reviewer 1 Report

Dear authors!

As far as I can tell, the work was written by students. Therefore, for the future, the questions of the reviewer must be answered by letter; Q&A, for example

reviewer page 2 line 57 correct Kalidium caspicum (L.) Ung.-Sternb., Kalidium cuspidatum (Ung. Sternb.) Grubov., Kalidium foliatum (Pall.) Moq., Kalidium gracile Fenzl and Kalidium schren kianum Bunge ex Ung.- Sternb. .

response from the author K. cuspidatum (Ung. Sternb.) Grubov., K. foliatum (Pall.) Moq., K. gracile Fenzl and K. schren kianum Bunge ex Ung.-Sternb. or answer corrected.

The authors correct as written above.

Where is table S2? Why was it not presented?

In general, the work can be published.

Reviewer 2 Report

The authors carefully addressed all comments in the revised version. So, the MS can be accepted in current form.